**Data Availability Statement:** The data involved in the study will be provided upon request, as it could be potentially identifiable and contains patient

# Spectrum of clinical features and genetic variants in mevalonate kinase (MVK) gene of South Indian families suffering from Hyperimmunoglobulin D Syndrome

Geeta Madathil Govindaraj[1,2☯]*, Abhinav Jain[3,4☯], Geetha Peethambaran[1], Rahul C. Bhoyar[3], Shamsudheen Karuthedath Vellarikkal[3], Arvind Ganapati[5], Pulukool Sandhya[5¤], Athulya Edavazhippurath[1,6], Dhananjayan Dhanasooraj[6], Jayakrishnan Machinary Puthenpurayil[1], Krishnan Chakkiyar[1], Anushree Mishra[3], Arushi Batra[3,4], Anu Punnen[7], Sathish Kumar[7], Sridhar Sivasubbu[3,4], Vinod Scaria[3,4]*

1 Department of Pediatrics, Government Medical College Kozhikode, Kozhikode, India, 2 Department of Pediatrics, FPID Regional Diagnostic Centre, Government Medical College Kozhikode, Kozhikode, India, 3 CSIR-Institute of Genomics and Integrative Biology, New Delhi, Delhi, India, 4 Academy of Scientific and Innovative Research (AcSIR), New Delhi, Delhi, India, 5 Department of Rheumatology, Christian Medical College, Vellore, India, 6 Multidisciplinary Research Unit, Government College Kozhikode, Kozhikode, India, 7 Department of Pediatrics, Christian Medical College, Vellore, India

☯ These authors contributed equally to this work.
¤ Current address: Department of Rheumatology, St. Stephen's Hospital, Delhi, India
* vinods@igib.in (VS); geetakkumar@gmail.com (GMG)

## Abstract

Hyper-IgD syndrome (HIDS, OMIM #260920) is a rare autosomal recessive autoinflammatory disorder caused by pathogenic variants in the mevalonate kinase (*MVK*) gene. HIDS has an incidence of 1:50,000 to 1:5,000, and is thought to be prevalent mainly in northern Europe. Here, we report a case series of HIDS from India, which includes ten patients from six families who presented with a wide spectrum of clinical features such as recurrent fever, oral ulcers, rash, arthritis, recurrent diarrhea, hepatosplenomegaly, and high immunoglobulin levels. Using whole exome sequencing (WES) and/or Sanger capillary sequencing, we identified five distinct genetic variants in the *MVK* gene from nine patients belonging to six families. The variants were classified as pathogenic or likely pathogenic as per the American College of Medical Genetics and Genomics and the Association for Molecular Pathology (ACMG-AMP) guidelines for annotation of sequence variants. Over 70% of patients in the present study had two recurrent mutations in *MVK* gene i.e. a nonsynonymous variant p. V377I, popularly known as the 'Dutch mutation', along with a splicing variant c.226+2delT in a compound heterozygous form. Identity by descent analysis in two patients with the recurrent variants identified a 6.7 MB long haplotype suggesting a founder effect in the South Indian population. Our analysis suggests that a limited number of variants account for the majority of the patients with HIDS in South India. This has implications in clinical diagnosis, as well as in the development of cost-effective approaches for genetic diagnosis and

sensitive information. Access to the data could be requested by mailing to Dr. Jyoti Yadav (j. yadav@igib.res.in) who is the convener of the Institutional Human Ethics Committee of CSIR-IGIB.

**Funding:** The study was funded by Council of Scientific and Industrial Research (CSIR, India) through MLP1801 (RareGen-CSIR India) and Science and Engineering Research Board EMR/2016/006828/HS (PAC Health Sciences). The funders had no role in study design, data collection and analysis, decision to publish, or preparation of the manuscript.

**Competing interests:** The authors have declared that no competing interests exist.

screening. To our best knowledge, this is the first and most comprehensive case series of clinically and genetically characterized patients with HIDS from India.

## Introduction

Autoinflammatory disorders are rare disorders characterized by dysfunction of the innate immune system, resulting in unprovoked inflammation. Hyper-IgD syndrome (HIDS) is one of the several autoinflammatory disorders and has an autosomal recessive mode of inheritance caused by pathogenic variants in the mevalonate kinase (*MVK*) gene [1]. Clinical features of HIDS are similar to those of hereditary periodic fever syndromes, which include recurrent fever, lymphadenopathy, hepatomegaly, rash, arthritis, abdominal pain, and increased levels of biomarkers like serum amyloid A (SAA), serum IgD and C-Reactive Protein (CRP) [2]. The acronym for the Hyper-IgD syndrome, was derived from the increased level of immunoglobulin D (IgD) in all the patients in the first described case series [3], but elevated IgD is not diagnostic of HIDS [4]. A more common feature in patients with HIDS is the deficiency of mevalonate kinase. In addition to HIDS, mevalonate kinase deficiency (MKD) also causes a severe phenotype called mevalonic aciduria (MA). MA is associated with neurological features like developmental delay, cerebellar atrophy, ataxia, psychomotor retardation and dysmorphic features, apart from autoinflammatory symptoms The clinical presentation largely depends on the residual activity of mevalonate kinase ranging from virtually undetectable in MA to a range of detectable, but clearly deficient in HIDS and possibly resulting in premature death [5]. Mevalonate kinase phosphorylates mevalonic acid, which is required for the formation of non-sterol isoprenoids and cholesterol. This in turn is necessary for prenylation of proteins for protein [2]. Due to decreased level of mevalonate kinase, RhoA remains unprenylated and increased secretion of proinflammatory cytokines such as IL-1β results in the development of autoinflammatory symptoms [2,6].

Case series from across the world and Eurofever registry have reported less than 200 HIDS patients worldwide mainly belonging to Caucasian population [7–9]. It has been estimated that 75% of patients with HIDS were from Western Europe i.e. the Netherlands, France, and Italy [10–12], whereas only a small number of patients with HIDS have been reported from Asian countries [2,13–17]. The prevalence of HIDS has been estimated to be in the range of 1:50,000 to 1:5000 [10]. The most common variants implicated in HIDS are p.V377I and p.I268T in the *MVK* gene. The variants typically occur in a compound heterozygous state in most of the patients [9,11,18,19]. The carrier frequency in the Dutch population is estimated to be 1 in 65. The frequency of V377I alleles within Dutch patients diagnosed with mevalonate kinase deficiency is 42% [10].

In this report, we describe a case series of ten patients with HIDS from six South Indian families identified by clinical suspicion and confirmed in nine of the patients with genetic testing. A majority of patients have two recurrent variants i.e. p.V377I and splice mutation 226 +2delT in the *MVK* gene in a compound heterozygous state. Analysis suggests a founder effect for these variants in the South Indian population. To the best of our knowledge, this is the first and most comprehensive report of patients with HIDS from India.

## Materials and methods

### Patients and clinical workup

The study has been approved by Government Medical College, Kozhikode by institutional ethical approval (Ethics No. GMCKKD/RP2017/IEC/147). Patients with suspected

autoinflammatory syndromes presented at two tertiary care hospitals in South India i.e. Government Medical College, Kozhikode, Kerala and Christian Medical College, Vellore, Tamil Nadu were taken up for systematic genetic analysis. Clinical characteristics of the children including age at onset, duration and interval between flares, presence of rash, lymphadenitis, arthritis, abdominal pain, diarrhea and hepatosplenomegaly were recorded in a semi-structured proforma. Inflammatory markers, total and differential counts were recorded during flares and immunoglobulin assays including IgD levels were performed. Whole exome sequencing for these patients was done at CSIR IGIB after a referral from the clinicians at the two hospitals with institutional ethical approval (Ethics No. GMCKKD/RP2017/IEC/147).

## DNA isolation and whole exome sequencing

After informed consent, approximately 5 ml of blood samples were drawn by venipuncture in acid citrate dextrose (ACD) tubes (Becton Dickinson, NJ, USA). Patient genomic DNA was isolated using a salting out method [20] and diluted to 100 ng as a template to perform WES. WES was performed on the patient samples using the Truseq Exome library prep kit followed the standard procedure as instructed by the manufacturer (Cat no.: 20000408, Illumina Inc., SA USA). The libraries were further sequenced on HiSeq2500/NovaSeq6000 platforms in a paired-end mode with read lengths of 150bp and estimated 100x coverage.

## Whole exome sequencing analysis

The raw reads were subjected to adapter and base quality trimming at a Phred score Q20 using Trimmomatic-0.38 [21]. The trimmed reads were mapped on the human reference genome version hg19/GRCh37 using Burrows-Wheeler Aligner (BWA) version 0.7.17, which aligns using a hash table based algorithm for fast and precise alignment [22]. The alignments were further sorted and duplicate reads were removed using SAMtools (version 0.1.19) [23] and Picard (version 2.21.1), [24] respectively. Finally, Genome Analysis ToolKit (GATK) best practices were used for variant calling using GATK (version 3.8.0) tool HaplotypeCaller [25–27].

## Variant annotation

The variants called were further systematically annotated using ANNOVAR (v.Date:2014-11-12 04:42:02), [28] which provides annotation from multiple databases such as RefGene [29], dbSNP (dbsnp152) [30] etc. Additionally, allele frequency was also annotated using global databases i.e. 1000 Genome Project (1000g2015aug_all), [31] which comprised of whole genome sequence data of 2504 healthy individuals from 26 populations, ESP6500 (esp6500si-v2_all), [32] which comprised of 6503 whole exome data from 2203 African-American and 4300 European-American individuals, and ExAC (exac03), [33] which comprised of exome data of 60,706 unrelated individuals who have European, African, South Asian, East Asian, and admixed American (hereafter referred to as Latino) ancestry. We also added the clinical significance annotation of variants from ClinVar v.20180603 [34].

## Variant prioritization

The variants were prioritized based on their potential to alter the protein structure or function (eg. missense, stop gain, stop loss, frameshift, non-frameshift, splicing, small indels, small insertion, and small deletion). These variants were further filtered to exclude variants whose minor allele frequency was greater than 5% in any of the population datasets due to the autosomal recessive mode of inheritance in the families. Further, variants were filtered based on predictions of their role in causing a disease using in-silico tools such as Sorting Intolerant From

Tolerant (SIFT) [35], Polymorphism Phenotyping v2 (Polyphen-2) [36] and Combined Annotation Dependent Depletion (CADD) [37]. The variants present in 320 genes that are known to cause primary immunodeficiency (PID) [38] and cataloged by experts of Inborn Errors of Immunity Committee were further filtered. The variants were further examined manually, and classified. Alignments for the variants were manually visualised on the IGV browser.

## Reclassification of variants as per the ACMG-AMP guidelines

The prioritized variants were manually classified as per 28 criteria for interpretation of sequence variants released by the American College of Medical Genetics and Genomics (ACMG) and the Association for Molecular Pathology (AMP). According to the guidelines, each variant was classified in one of the five categories—pathogenic, likely pathogenic, variant of uncertain significance (VUS), benign, or likely benign based on the evidence from sequence annotation, minor allele frequency in population datasets, computational predictions, segregation in families, and other databases according to the algorithm provided by the ACMG and AMP experts [39].

## Variant validation using Sanger sequencing

Oligonucleotide primers for polymerase chain reaction (PCR) were designed to sequence pathogenic and likely pathogenic variants in the probands and their family members. We also performed Sanger sequencing in the hotspot region for the patient P6 who had not undergone WES. The primer sequences used for amplicon amplification and conditions are detailed in S1 File. The amplified products were purified using QIAquick PCR purification (Qiagen, Maryland, USA) according to the manufacturer's instructions and capillary sequencing was performed using BigDye Terminator chemistry (Applied Biosystem, California USA).

## Global screening array

DNA from 24 samples was used for performing genotyping using the Global Screening Array (GSA) (Illumina, USA) as per standard protocols provided by the manufacturer. Briefly, DNA is denatured and an isothermal amplification was performed overnight at 37 degrees. Amplified DNA is further sheared and precipitated by centrifugation at 4 degrees. The genotypes were called using the genotype calling module of GenomeStudio software (Illumina, USA) following the standard protocol.

## Identity by descent

Identity by descent estimates the genetic similarity among individuals by inheritance. The pruned genotype data from 24 rare disease patient samples was used to perform multiple analysis depending on the type of disease. The 24 samples also included our patients P1 and P7, on whom we performed IBD analysis using Kinship-based INference for Genome-wide association studies (KING ver. 2.1.4 http://people.virginia.edu/~wc9c/KING) [40]. KING predicts up to 4th degree of relationship among individuals or the IBD segment which is shared among individuals. The IBD plots for patients P1 and P7 were plotted using KING.

# Results

## Clinical presentation

A total of ten patients from South India suspected to have autoinflammatory diseases were included in the series. Of these, eight patients were referred for genetic evaluation from the Government Medical College, Kozhikode, Kerala and two from the Christian Medical College,

Vellore, Tamil Nadu. The Clinical features of these patients are well tabulated in S1 Table. All patients had recurrent episodes of fever with inflammatory manifestations of the skin, mucosae, joints and serosal surfaces with elevated acute phase reactants in the absence of infection and with no evidence of autoantibodies. There were five girls and five boys who belonged to six families, and included four pairs of siblings. Pedigree of families is depicted in S1 Fig.

All children were symptomatic before the age of sixteen months. 80% of the patients had their first symptoms during infancy, while two children had a recurrent fever in the neonatal period. All children had recurrent episodes of fever, diarrhea, hepatosplenomegaly and rash, 80% had cervical adenopathy during flares, 60% had arthritis and oral ulcers, and 60% reported that flares were triggered following vaccination. The manifestations during flares were inconsistent, some being associated with a rash, others with oral ulcers and so on. The rash varied from macular to urticarial.

Four children had bacterial infections that included pneumonia, periapical dental abscess, meningitis, septic shock and urinary tract infection. One child had recurrent molluscum contagiosum. Four of the children needed surgical procedures for choledochal cyst, low anorectal malformation, undescended testes and perianal fistula, one child required adhesiotomy for intestinal obstruction.

Apart from elevated acute phase reactants during flares, a notable feature was the elevated levels of IgA in all children and elevated IgE levels in most children for whom IgE level information was available. Serum B12 levels were elevated in seven children for whom reports were available.

The diagnostic possibilities which had been considered previously in these patients included the autoimmune lymphoproliferative syndrome, lymphoma, systemic-onset juvenile idiopathic arthritis, Castleman Disease, histiocytosis, autoinflammatory and immunodeficiency syndromes.

Biologicals for IL-1 blockade were not used due to its unavailability and flares were managed with NSAIDs and corticosteroids. As our patients had increased risk of infections, they were commenced on cotrimoxazole prophylaxis. 75% of children showed improvement over time eg. P3 was hospitalized 4–5 times each below 5 years and between 5 and 10 years respectively but did not require admission after 10 years of age. Similarly, P4 was admitted 5–6 times below the age of 3 years but did not need admission after 6 years of age. Another reason for improvement could be cotrimoxazole prophylaxis which seemed to result in reduced frequency of febrile episodes in 80% of those in whom it was prescribed. The mortality was 10%. One child succumbed while at home after recovering from a respiratory infection for which she had been hospitalized. The cause of death could not be ascertained.

## Whole exome sequencing

Whole exome sequencing was performed for five patients (P1, P2, P4, P7, and P9 from families F1, F2, F3, F5, and F6, respectively) provisionally diagnosed with an autoinflammatory disorder. An average of 63.3 million raw reads were generated for each patient. Raw reads were trimmed using Trimmomatic-0.38 and filtered reads were aligned to the human reference genome hg19/GRCh37 using BWA 0.7.17. On average 45.8 million reads for each patient were aligned on the human reference genome hg19/GRCh37 with the average mapping of 99% and average coverage of over 175X. Variant calling was done using GATK HaplotypeCaller, which generates on average 554,832 variants for each patient. Details of the data generated by the whole exome analysis for each patient has been well tabulated in S2 Table.

## Variant filtering

Non-coding variants such as intergenic, intronic, ncRNA intronic, ncRNA exonic, etc, as well as synonymous SNV were filtered out. The average filtered variants were 12,661 for each patient. These variants were further filtered for minor allele frequency (MAF) < 5% in 1000 Genomes Project, ESP6500si and ExAC that brought down the average number of variants to 1,918 for each patient. We prioritized an average of 27 genetic variants that were present in the 320 primary immunodeficiency genes in each patient for further downstream analysis. Variants were further prioritized based on the phenotype-genotype correlation that mainly has compound heterozygous inheritance that led to 2, 2, 1, 2, and 2 variants in the *MVK* gene for P1, P2, P4, P7, and P9 patients respectively. The prioritized variants that could lead to HIDS in patients after at each filtering step have been tabulated in the S2 Table. Since HIDS is an autosomal recessive disorder and typically caused by homozygous or compound heterozygous variants, we further performed Sanger capillary sequencing to examine whether compound heterozygous variants were in cis or trans alignment with each other and also to validate WES results.

## Variant validation using Sanger capillary sequencing

We performed PCR amplification using the primers as detailed previously followed by Sanger capillary sequencing for all patients who had undergone exome sequencing. For P6, we did not have the parents' DNA samples. We could not validate the variants in P8 of family 5 as we did not have an adequate DNA sample. However, since P7 and P8 belong to the same family and manifested similar clinical features, we hypothesized that P8 also inherited the same genetic mutation as P7. In other patients, we found variants were either trans compound heterozygous or homozygous segregation in the family. Sanger capillary sequencing results are represented for each of the patients and families in S1 Fig. The variants are also tabulated in Table 1. The linear gene and 3-dimensional protein visualization with the mutation sites by PyMOL (ver 2.4) [41] have been represented in Fig 1.

## Classification of prioritized variants as per the ACMG/AMP guidelines

All the variants were systematically evaluated as per the 28 parameters in the ACMG/AMP guidelines. All the patients affected by HIDS had pathogenic or likely pathogenic mutations in *MVK* gene as per the ACMG/AMP guidelines. We found two pathogenic mutations c.226 +2delT and p.V377I which were recurring in a compound heterozygous state (trans configuration) in our patients. Five patients (P1, P6, P7, P9, and P10) carried p.V377I mutation in a compound heterozygous state with splice site mutation c.226+2delT in four patients (P1, P7, P9, and P10) whereas P6 carried another nonsynonymous mutation p.E193Q in a compound heterozygous state and classified as likely pathogenic. The splicing mutation c.226+2delT was also found in a trans compound heterozygous state with p.N205D in two siblings P2 and P3 of family 2, which was classified as likely pathogenic. A homozygous nonsynonymous mutation i.e. D366G has been classified as likely pathogenic in patients P4 and P5 of family 3, in which there was a third degree consanguineous parentage. The prioritized variants along with their ACMG annotation have been tabulated in Table 1.

## Identity by descent analysis

The genotyping dataset generated on the Global screening array had a genotype call rate of ~99%. In both of our patients P1 and P7, we could not find any mutual blood relationship up to 4th degree, but we found both of them shared the haplotype that spans the genomic loci of

**Table 1. ACMG classification of prioritized variants in patients with HIDS.**

| Patient | Family | Gene | Chr:pos | Variant | SNPid | Coding change | ACMG Criterion | ACMG Classification | Genotype |
|---|---|---|---|---|---|---|---|---|---|
| P1 | F1 | MVK | chr12:110013951 | GT>G | . | c.226+2delT | PVS1, PM2, PM3 | Pathogenic (1b) | Compound Heterozygous |
|  |  | MVK | chr12:110034320 | G>A | rs28934897 | c.1129G>A:p.V377I | PS3, PS4, PP1, PP5, BP4 | Pathogenic (II) |  |
| P2 | F2 | MVK | chr12:110013951 | GT>G | . | c.226+2delT | PVS1, PM2, PM3 | Pathogenic (1b) | Compound Heterozygous |
|  |  | MVK | chr12:110023912 | A>G | rs104895364 | c.613A>G: p.N205D | PM1, PM2, PP1, PP3 | Likely pathogenic (V) |  |
| P3 | F2 | MVK | chr12:110013951 | GT>G | . | c.226+2delT | PVS1, PM2, PM3 | Pathogenic (1b) | Compound Heterozygous |
|  |  | MVK | chr12:110023912 | A>G | rs104895364 | c.613A>G: p.N205D | PM1, PM2, PP1, PP3 | Likely pathogenic (V) |  |
| P4 | F3 | MVK | chr12:110034288 | A>G | . | c.1097A>G:p.D366G | PM1, PM2, PP1, PP3 (Yellow) | Likely pathogenic (V) | Homozygous |
| P5 | F3 | MVK | chr12:110034288 | A>G | . | c.1097A>G:p.D366G | PM1, PM2, PP1, PP3 | Likely pathogenic (V) | Homozygous |
| P6 | F4 | MVK | chr12:110034320 | G>A | rs28934897 | c.1129G>A:p.V377I | PS3, PS4, PP1, PP5, BP4 | Pathogenic (II) | Compound Heterozygous |
|  |  | MVK | chr12:110023876 | G>C | . | c.577G>C: p.E193Q | PM1, PM2, PM3, PP3 | Likely pathogenic (IV) |  |
| P7 | F5 | MVK | chr12:110013951 | GT>G | . | c.226+2delT | PVS1, PM1, PM3 | Pathogenic (1b) | Compound Heterozygous |
|  |  | MVK | chr12:110034320 | G>A | rs28934897 | c.1129G>A:p.V377I | PS3, PS4, PP1, PP5, BP4 | Pathogenic (II) |  |
| P9 | F6 | MVK | chr12:110013951 | GT>G | . | c.226+2delT | PVS1, PM2, PM3 | Pathogenic (1b) | Compound Heterozygous |
|  |  | MVK | chr12:110034320 | G>A | rs28934897 | c.1129G>A:p.V377I | PS3, PS4, PP1, PP5, BP4 | Pathogenic (II) |  |
| P10 | F6 | MVK | chr12:110013951 | GT>G | . | c.226+2delT | PVS1, PM2, PM3 | Pathogenic (1b) | Compound Heterozygous |
|  |  | MVK | chr12:110034320 | G>A | rs28934897 | c.1129G>A:p.V377I | PS3, PS4, PP1, PP5, BP4 | Pathogenic (II) |  |

6.7 megabases (MB) at chr12:107138141–114000478 and consists of 1407 Single Nucleotide Variants (SNVs) on the array. The genomic loci overlapped with the recurrent variants (p. V377I and c.226+2delT). The shared haplotype between two patients at the genomic level in different chromosome has been well represented in Fig 2.

## Discussion

In the present report, we describe one of the largest case series of patients with HIDS from South India, which provides a unique view of the clinical features. There were four pairs of affected siblings, who had a similar spectrum of symptoms, but varied in their age at presentation and severity. We compared clinical features and genetic mutation of our cohort to the cohorts which have more than five patients reported in different populations which include Italian [12,42,43], French and Belgian [44], Dutch [45], Japanese [46], German [47], and other cohorts, which mainly represent European patients [7,8]. The comparison of patients with HIDS of different populations with our study has been well represented in Table 2.

Notably, age at disease onset varied widely among different populations. In this regard, South Indian patients were similar to the European population. However, oral ulcers, myalgia and splenomegaly were significantly lower in the cohort of European patients in comparison to South Indian. As reported in previous studies of the European cohorts, 80% of children were symptomatic from infancy, and two children had a neonatal-onset [48]. Rash, diarrhoea, hepatosplenomegaly were seen in 100% of our patients. So in a patient with the triad of

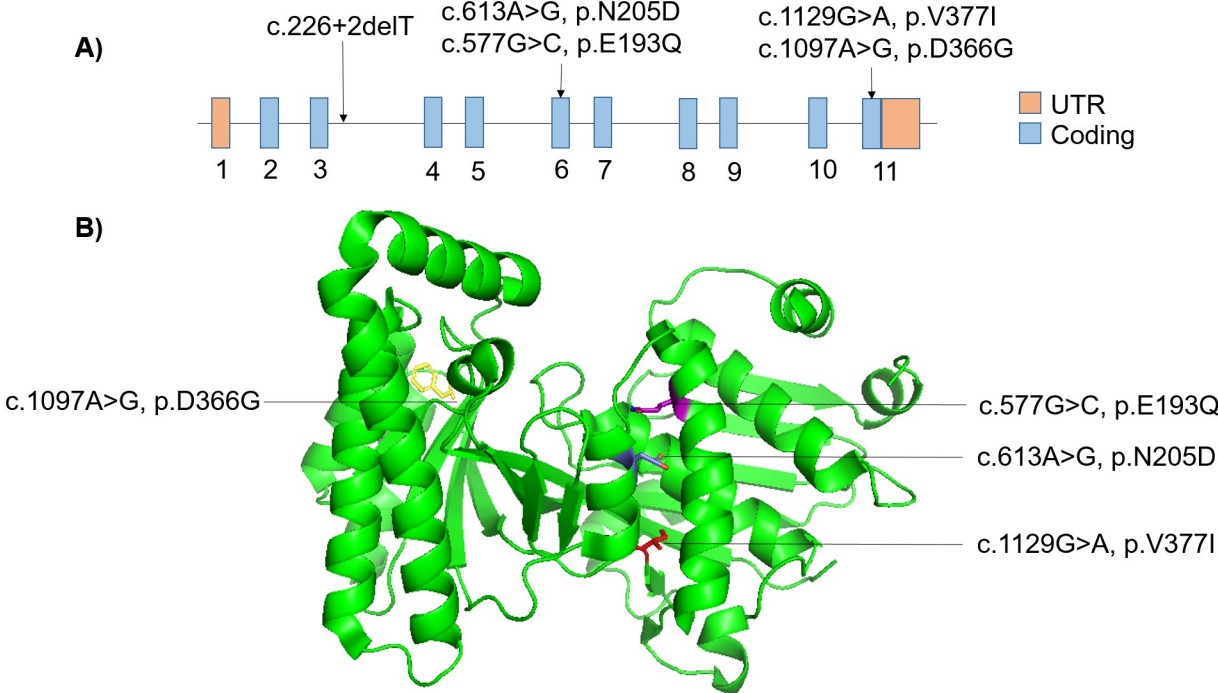

**Fig 1. Mutation in the *MVK* gene in linear and in 3-dimensional Mvk protein visualization.** A) Linear model of *MVK* gene with a mutation associated with Hyper-IgD syndrome in South Indian patients. B) 3-dimensional Mvk protein visualization with the mutation sites associated with Hyper-IgD syndrome in South Indian patients.

symptoms and recurrent fever, diagnosis of HIDS should be strongly considered in south Indian patients.

An increased frequency of infections of possible bacterial etiology was observed in our patients, similar to French, Italian, and Belgian reports, which have indicated enhanced susceptibility to pneumococcal infections [8,12,44]. Elevated levels of mevalonic acid in the plasma due to deficient activity of mevalonate kinase have been postulated to favor the persistence of S. pneumoniae in lungs and serum. Immunization against pneumococcal infections is therefore strongly recommended in patients with HIDS. This is a pointer to the fact that HIDS is associated with immune deficiency apart from autoinflammation. The reduction in the number of flares in the majority of children in whom cotrimoxazole was started as antimicrobial prophylaxis is also a pointer in this direction, but requires more studies to confirm its benefit, since it was a subjective observation of the treating clinician. It would be a useful and simple modality that can be tried to reduce the frequency of flares, especially in regions where biologicals are either not readily available or where there are cost constraints.

Although HIDS is usually unassociated with impaired growth, 60% of children had low height for age, possibly due to recurrent and frequent inflammatory episodes, and 50% had mild motor developmental delay. Severe anemia requiring packed red cell transfusions was a feature in more than half of the children and is attributable to the frequent and severe inflammatory episodes. A common feature observed on follow up was improvement of symptoms with age, that has been previously described [49].

Due to the limited availability of IgD assay, it was possible to estimate levels in only 3 patients, whose IgD levels were elevated. Estimation of IgD levels as a pointer to the need to

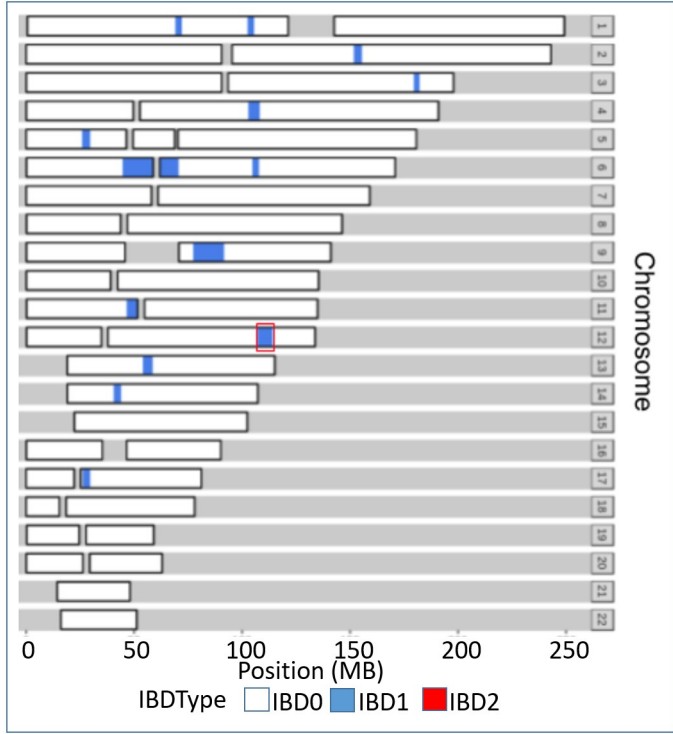

**Fig 2. Haplotype sharing among P1 and P7 patients at the genome level.** Shared segmental SNVs are represented in blue in different chromosomes of P1 and P7 patients. Red outline at chr12 of a shared haplotype among P1 and P7 composed of both pathogenic variants which were found to be recurrent in patients with HIDS from South India.

look for an *MVK* variant is not recommended due to its poor predictive value and specificity. Elevated IgD levels may be found in certain infections, malignancies, and inherited disorders and is now considered to be an epiphenomenon [4,50]. Elevated IgA levels have been reported to occur in HIDS, due to an increase in polymeric-IgA and was observed in all our patients. This is a useful diagnostic clue [51]. IgE levels were elevated in all children in whom it had been assayed, a finding that has not been reported earlier and would need further studies to be confirmed.

Several patients went through a diagnostic odyssey before whole exome sequencing revealed the diagnosis. Among our patients, 40% were diagnosed at or above the age of 10 years. The median time from onset to diagnosis has been reported to be 14 years in patients with no positive family history and as many as a third of patients with HIDS have been previously misdiagnosed [8]. This is because the rarity of the disease and consequent lack of familiarity with the treating physician mandates the exclusion of more common diseases upfront. The non—specific nature of symptoms is also a contributing factor. One child underwent laparotomy with adhesiotomy and appendicectomy. Confirmation of the diagnosis of HIDS also helps prevent unnecessary diagnostic and therapeutic procedures [52].

One child in our cohort succumbed to the disease while at home, and hence the exact cause could not be determined. Deaths of three patients between 2 and 3 years of age have been reported earlier due to multiple organ failure and macrophage activation syndrome with staphylococcal sepsis [8,44,53].

The new Eurofever / PRINTO classification criteria stipulate the presence of one among three clinical criteria including gastrointestinal symptoms, cervical lymphadenitis, and

**Table 2. Comparison of patients with HIDS of our cohort across different population groups.**

| Reference | Haar et al. 2016 [7] | Hilst et al. 2008 [8] | Doglio et al. 2013 [42] | Bader-Meunier et al. 2011 [44] | D'Osualdo et al. 2005 [12] | Frenkel et al. 2001 [45] | Tanaka et al. 2019 [46] | Stabile et al. 2013 [43] | Lainka et al. 2012 [47] | |
|---|---|---|---|---|---|---|---|---|---|---|
| Population | Mainly European | Mainly European | Italian | French and Belgian | Italian | Dutch | Japanese | Italian | German | Present study |
| Number of Patients | 114 | 103 | 56 | 50 | 15 | 14 | 10 | 10 | 8 | 10 |
| Gender | 53M 61F | 52M 51F | 29M 27F | 21M 29F | 9M 6F | 9M 5F | 4M 6F | 4M 6F | 4M 4F | 5M 5F |
| Average Age (years) | NA | 19 (2–74) | 13.3 ± 8.5 | 21.6 (1–58) | 10 (2–27) | 9.4 (2–20) | 8.3 (0.6–15.3) | NA | 7 (2–14) | 6.45 (1.5–15) |
| Disease onset (months) | 6 | 6 (0–10) | 10.5 ± 15.3 (1–108) | 24.5 (0.03–504) | 10 (1–24) | 15 (0–160) | 1 (0–3) | 42.6 (2–110) | 13.5 (12–24) | 6.23 (0.5–24) |
| Duration of fever (days) | 4 | NA | NA | 3.7 (1–10) | 4.5 (3–7) | 4 (3–7) | 4 to 7 | 5 | 5.5 (3–12) | 3–7 |
| Number of spikes/year | 11 to 15 | 7 to > 12 | 13.8 ± 5.4 (3–30) | NA | 12.5 (10–18) | 16 (6–26) | 7 to 11 | NA | 4–52 | 20 (12–24) |
| Ulcers n/m (%) | 18/114 (16) | 50/103 (48.5) | NA | 8/49 (16.3) | NA | NA | NA | 6/10 (60) | 2/8 (25) | 6/10 (60) |
| Chills n/m (%) | NA | 65/103 (62.7) | NA | NA | 6/15 (40) | 6/14 (35.7) | NA | NA | NA | NA |
| Fever on vaccination n/m (%) | 38/99 (38.3) | NA | NA | NA | NA | 13/14 (92.8) | NA | NA | NA | 4/10 (40) |
| Abdominal pain n/m (%) | 98/111 (88.2) | 88/103 (85.4) | NA | 30/49 (61.2) | 15/15 (100) | 12/14 (85.7) | 4/10 (40) | 9/10 (90) | 8/8 (100) | 4/10 (40) |
| Diarrhoea n/m (%) | 93/111 (83.7) | 74/103 (71.6) | NA | 34/49 (69.3) | 12/15 (80) | 12/14 (85.7) | 6/10 (60) | NA | 6/8 (75) | 10/10 (100) |
| Vomiting n/m (%) | 76/110 (69) | 73/103 (70.9) | NA | 21/49 (42.8) | NA | 11/14 (78.5) | 6/10 (60) | NA | 6/8 (75) | 3/10 (30) |
| Arthralgia n/m (%) | 80/113 (70.7) | 86/103 (83.5) | NA | 33/49 (67.3) | 12/15 (80) | 12/14 (85.7) | 2/10 (20) | 9/10 (90) | 4/8 (50) | NA |
| Arthritis n/m (%) | 31/109 (28.4) | 57/103 (55.3) | NA | 20/49 (40.8) | NA | 4/14 (28.5) | 2/10 (20) | NA | NA | 6/10 (60) |
| Myalgia n/m (%) | 64/112 (57.1) | NA | NA | 10/49 (20.4) | 9/15 (60) | NA | NA | NA | 2/8 (25) | 6/8 (75) |
| Rash n/m (%) | 43/111 (38.7) | NA | NA | 23/35 (65.7) | 7/15 (46.6) | 11/14 (78.5) | 9/10 (90) | 5/10 (50) | 5/8 (62.5) | 9/10 (90) |
| Lymphadenopathy n/m (%) | 96/113 (84.9) | 90/103 (87.4) | NA | 35/49 (71.4) | 15/15 (100) | 12/14 (85.7) | 7/10 (70) | 10/10 (100) | 7/8 (87.5) | 8/10 (80) |
| Splenomegaly n/m (%) | NA | 34/103 (32.4) | NA | 31/49 (63.2) | 8/15 (53.3) | 7/14 (50) | 3/10 (30) | NA | NA | 10/10 (100) |
| MVK mutation n/m (%) | 114/114 (100) | 103/103 (100) | 100 | 49/50 (98) | 15/15 (100) | 11/14 (78.5) | 5/10 (50) | 7/10 (70) | 8/8 (100) | 9/10 (90) |
| Recurrent Mutation V377I n/m (%) | 96/114 (84.2) | 52/103 (50) | 42/56 (75) | 36/49 (73) | 12/15 (80) | 11/11 (100) | NA | 2/7 (28.5) | 6/8 (75) | 5/10 (50) |
| Second Recurrent Mutation n/m (%) | 29/114 (25.4) I268T | 15/103 (14.7) I268T | NA | 8/49 (16.3) I268T | 4/15 (26.6) I268T | 4/11 (36.4) I268T | NA | 5/7 (71.4) I268T | 2/8 (25) I268T, 2/8 (25) R277H | 6/10 (60) c.226+2T |

NA: Not Applicable.

aphthous stomatitis along with a confirmatory mutation in the *MVK* gene, which was satisfied by all the patients except P8 due to unavailability of the sample [54]. We hypothesize that P8 will have the same mutation as P7, as both of them are siblings with similar clinical features.

Genetically, South Indian patients have two recurrent mutations, p.V377I and c.226+2delT, where V377I has been widely reported in all the populations except Japanese. The c.226+2delT is a unique mutation found exclusively in the Indian subcontinent, only been reported once previously in South Indian patients [13]. All the patients were found to carry pathogenic or likely pathogenic mutations in *MVK* gene described previously in HIDS either in a compound heterozygous or homozygous state. Out of nine patients genetically evaluated in our study, five patients from four unrelated families had V377I mutation (rs28934897) along with splice site mutation c.226+2delT and one patient had the splice variant along with the with non synonymous mutation p.E193Q. The V377I (rs28934897), is also popularly known as the 'Dutch variant' as its carrier frequency in Dutch population is quite high i.e. 1:153 individuals. Patients with HIDS had V377I mutation in homozygous or trans compound heterozygous states around the world have a similar phenotype [7,55]. It is also prevalent in other parts of the globe which involves Italy, France, Belgium, Germany, Spain, and Arabic countries [6–10,12,42–44,47,56–63]. It has low allele frequency in the control population i.e. ~0.001 in both gnomAD and Esp6500, and is absent in 1000 Genome Project. In-silico tools predicted it as tolerated by SIFT and benign by PolyPhen, where CADD score > 20 which predicted it as disease causing.

The second recurrent variant is a splice mutation c.226+2delT in *MVK* gene in six patients from four unrelated families. This mutation has been reported once in a South Indian family who was also from Kerala, India inherited in a compound heterozygous state with p.V377I [13] as in four patients (P1, P7, P9, and P10) from three unrelated families. It is absent in the control populations i.e. 1000 Genome Project, gnomAD, and Esp6500. This mutation may lead to the retention of intron 3 or skipping of exon 4 in the complementary DNA which in turn affects mRNA stability or protein activity. There are reports of splicing mutation, 2bp upstream to our mutation i.e. c.227-1G>A that resulted in exon skipping and affects protein activity [14,64]. Other nonsynonymous variations described in this case series are N205D (rs104895364), E193Q and D366G. N205D mutation has been reported around the globe in trans compound heterozygous manner with different mutation worldwide including p.V377I from Turkey [65], p.K13Q from France [66], Y114fs from the Netherlands [55], and c.382_383 delAG* from Japan [46]. N205D has very low allele frequency in the control population i.e. 3.18e-05 in gnomAD, and 4.99e-05 in Esp6500, and is absent in 1000 Genome Project. It falls in the GHMP kinase domain of MVK protein, and affects the protein folding [67]. Computational tools (SIFT, PolyPhen, and CADD) predicted it as pathogenic. Similarly p.E193Q has been predicted as pathogenic by computational tools and it falls in the GHMP kinase domain of MVK protein, which disrupts the catalytic and binding of the MVK functionally shown in rats and 293T cells [67,68]. However, there is only one prior report of p.D366G homozygous mutation and this was again from Kerala, India [69]. It is absent in the control population databases, and predicted as pathogenic by SIFT and CADD, whereas benign by PolyPhen. It falls in the GHMP kinase domain of MVK.

Patients P1, P7, P9, and P10 had the same variants c.226+2delT and p.V377I with similar clinical characteristics, but with variable severity, the duration and frequency of flares also differed. p.N205D variant was found to be associated with the occurrence of recurrent arthritis. Painful episodes of cervical lymphadenitis were a prominent feature in the two siblings who were homozygous for the p.D366G variant. Lymphoproliferation was marked in children with the splicing variant.

In this study, we considered WES, in contrast to targeted sequencing of MVK gene, due to the fact that targeted sequencing was not widely available, was expensive and time consuming. The study also enabled development of a targeted capillary sequencing based approach encompassing the prevalent variants in the population, because a genetic diagnosis is not only essential for diagnosis, but also for genetic counseling, and for screening and diagnosis of related members of the family.

In summary, the present report on patients with HIDS emphasizes the variability of clinical presentation and the genetically heterogeneous nature of the disease. We found two recurrent mutations from South India which were inherited in the trans compound heterozygous states could be considered as potential founder MVK variants in the South Indian populations. However, considering the variability and overlapping clinical features, clinical diagnosis of HIDS may not be possible in all cases as seen in this case series. When the autoinflammatory disease is suspected, WES must be considered unless the clinical features and other labs are very typical of HIDS. In a South Indian patient who manifests autoinflammatory features typical of HIDS Sanger sequencing for V377I and c.226+2delT in *MVK* gene, which are the two most common mutations described could be considered.

## Supporting information

**S1 Fig. Pedigrees and chromatograms for the pathogenic/likely pathogenic variants typed in the family using Sanger capillary sequencing.** Samples marked $-were analysed using whole exome sequencing and marked # were analysed using Sanger capillary sequencing. (TIF)

**S1 Table. Clinical features of patients with HIDS in the present cohort.** (XLSX)

**S2 Table. Data summary of exome sequencing for five patients with HIDS.** (XLSX)

**S1 File. Primer sequence for PCR amplification and Sanger sequencing with coordinates (GRCh37 or hg19).** (DOCX)

## Acknowledgments

Authors acknowledge Bani Jolly, Disha Sharma, and Mukta Poojary for suggestions, which enriched the manuscript.

## Author Contributions

**Conceptualization:** Geeta Madathil Govindaraj, Sridhar Sivasubbu, Vinod Scaria.

**Data curation:** Geeta Madathil Govindaraj, Abhinav Jain, Geetha Peethambaran, Pulukool Sandhya, Athulya Edavazhippurath, Dhananjayan Dhanasooraj, Jayakrishnan Machinary Puthenpurayil, Anushree Mishra.

**Formal analysis:** Geeta Madathil Govindaraj, Abhinav Jain, Rahul C. Bhoyar, Shamsudheen Karuthedath Vellarikkal, Arvind Ganapati, Pulukool Sandhya, Athulya Edavazhippurath, Dhananjayan Dhanasooraj, Jayakrishnan Machinary Puthenpurayil, Krishnan Chakkiyar, Anushree Mishra, Arushi Batra, Anu Punnen, Sathish Kumar.

**Funding acquisition:** Sridhar Sivasubbu.

**Investigation:** Geeta Madathil Govindaraj, Abhinav Jain, Geetha Peethambaran, Shamsudheen Karuthedath Vellarikkal, Pulukool Sandhya, Athulya Edavazhippurath, Dhananjayan Dhanasooraj, Jayakrishnan Machinary Puthenpurayil, Krishnan Chakkiyar.

**Methodology:** Geeta Madathil Govindaraj, Abhinav Jain, Geetha Peethambaran, Shamsudheen Karuthedath Vellarikkal, Sridhar Sivasubbu.

**Project administration:** Vinod Scaria.

**Resources:** Geeta Madathil Govindaraj, Sridhar Sivasubbu, Vinod Scaria.

**Supervision:** Geeta Madathil Govindaraj, Pulukool Sandhya, Sridhar Sivasubbu, Vinod Scaria.

**Validation:** Abhinav Jain.

**Visualization:** Abhinav Jain.

**Writing – original draft:** Geeta Madathil Govindaraj, Abhinav Jain, Pulukool Sandhya, Vinod Scaria.

**Writing – review & editing:** Geeta Madathil Govindaraj, Abhinav Jain, Pulukool Sandhya, Sridhar Sivasubbu, Vinod Scaria.

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
