## [Decision Letter · Decision Letter 0]

19 May 2020

PONE-D-20-12540

Spectrum of clinical features and genetic variants in mevalonate kinase (MVK) gene of Indian families suffering from Hyperimmunoglobulin D Syndrome

PLOS ONE

Dear Dr. Scaria,

Thank you for submitting your manuscript to PLOS ONE. After careful consideration, we feel that it has merit but does not fully meet PLOS ONE’s publication criteria as it currently stands. Therefore, we invite you to submit a revised version of the manuscript that addresses the points raised during the review process.

We would appreciate receiving your revised manuscript by Jul 03 2020 11:59PM. To enhance the reproducibility of your results, we recommend that if applicable you deposit your laboratory protocols in protocols.io, where a protocol can be assigned its own identifier (DOI) such that it can be cited independently in the future. For instructions see: http://journals.plos.org/plosone/s/submission-guidelines#loc-laboratory-protocols

We look forward to receiving your revised manuscript.

Kind regards,

Dror Sharon, PhD

Academic Editor

PLOS ONE

Journal Requirements:

Additional Editor Comments (if provided):

Reviewers' comments:

Reviewer's Responses to Questions

**Comments to the Author**

1. Is the manuscript technically sound, and do the data support the conclusions?

Reviewer #1: Yes

Reviewer #2: Yes

2. Has the statistical analysis been performed appropriately and rigorously? 

Reviewer #1: N/A

Reviewer #2: N/A

3. Have the authors made all data underlying the findings in their manuscript fully available?

Reviewer #1: Yes

Reviewer #2: Yes

4. Is the manuscript presented in an intelligible fashion and written in standard English?

Reviewer #1: Yes

Reviewer #2: Yes

5. Review Comments to the Author

Reviewer #1: This is a valuable contribution to the literature on HIDS (Hyper IgD Syndrome) or MVK deficiency. By presenting a small but by HIDS standard large case series with both clinical and molecular data it is a very useful contribution. The ten patients from six families represent the clinical spectrum previously reported with the additional information on the difficulty making the correct diagnosis and the other disorders that had been entertained before the molecular studies clearly identified the proper defect. Virtually nothing was said about outcome or treatment perhaps because the standard therapies that block IL1 (anakinra and canakinumab) are apparently unavailable or they plan to write another paper with that data. Surprisingly the authors state that the patients did well whereas untreated HIDS is typically associated with poor growth and development. None of the patients had the more severe form of the disease called mevalonic aciduria which is associated with severe and life threatening manifestations in infancy but not the periodic fever syndrome that HIDS displays.

I liked tables 1 and 2 but not Figure 1 which I would like to replace with a linear model of the gene with the mutation sites indicated. Another figure which would be useful is a three dimensional model of the enyme with the mutation sites identified. A little more focussed discussion of the deletion mutation and its effect on enzyme function or enzyme stability would have been interesting. The sequencing data is consistent with autosomal recessive inheritance requiring either homozygosity or compound heterozygote suggesting single mutations even of the deletion variant have enough residual activity to not be clinically symptomatic.

Reviewer #2: Thank you for the opportunity to review this manuscript. Authors have desribed the clinical and molecular spectrum of Hyperimmunoglobulin D Syndrome. I have few comments-

Introduction-

Second para- 4th line- Authors could add more Indian refernces on MVK disorder. Some of the important references are as follows-

1: Correa ARE, Gupta N, Bagri N, Vignesh P, Alam S, Yamaguchi S. Mevalonate

Kinase Deficiency as Cause of Periodic Fever in Two Siblings. Indian Pediatr.

2020 Feb 15;57(2):180-181. PubMed PMID: 32060250

2: Jain A, Misra DP, Sharma A, Wakhlu A, Agarwal V, Negi VS. Vasculitis and

vasculitis-like manifestations in monogenic autoinflammatory syndromes. Rheumatol

Int. 2018 Jan;38(1):13-24. doi: 10.1007/s00296-017-3839-6. Epub 2017 Oct 14.

Review. PubMed PMID: 29032440.

3: Abdulla MC, Hawkins PN, Alungal J, Rowczenio D. Hyperimmunoglobulin D syndrome

in an Indian family undiagnosed for 11 years. Int J Rheum Dis. 2017

Dec;20(12):2236-2237. doi: 10.1111/1756-185X.12811. Epub 2015 Dec 1. PubMed PMID:

26620804.

4: Sinha A, Waterham HR, Sreedhar KV, Jain V. Novel mutations causing

hyperimmunoglobulin D and periodic fever syndrome. Indian Pediatr. 2012

Jul;49(7):583-5. PubMed PMID: 22885443.

Methods-

Clinical work up- Wheteher any biochemical assays like GCMS was performed ?In how many a clinical diagnosis of Hyper IgD was made in the absence of igD levels.

Variant prortization- Generally a MAF of 1% is used. The reason behind using MAF of 5%is may be specified.

Table 1- Elevated IgD is seen in patient 1 only.was it performed in one patient ? or was it normal in others. I understand that it is not widely available.

Information on consanguinity should be added in the table.

Highlight the significance of elevated Vit B12

Discussion-

Did authors find any genotype pheotype correlation

Authors have stated that p.V377I and c.226+2delT are the common mutations in Indian subcontinent. This should be stated after incorporating the complete Indian data.

Was there any analysis done to prove the founder effect?

I think this line at the end of discussion ("In a patient who manifests

......... for V377I and c.226+2delT in MVK)may not be generalisable as Indian population is very heterogenous and this study mainly included the south Indian population .

General comments-

Minor typos in intorduction and discussion

6. PLOS authors have the option to publish the peer review history of their article (what does this mean?). If published, this will include your full peer review and any attached files.

Reviewer #1: No

Reviewer #2: No

---

## [Author Response · Author response to Decision Letter 0]

9 Jul 2020

Journal Requirements:

Response: Thanks for pointing that out, we have made the changes according to the PLOS ONE’s style.

Response: We apologize for not making it clear. We have updated the cover letter with the information of data availability on request by the IGIB Institutional Ethics committee with their contact details.

Comments to the Author

1. Is the manuscript technically sound, and does the data support the conclusions?

Reviewer #1: Yes

Reviewer #2: Yes

2. Has the statistical analysis been performed appropriately and rigorously?

Reviewer #1: N/A

Reviewer #2: N/A

3. Have the authors made all data underlying the findings in their manuscript fully available?

Reviewer #1: Yes

Reviewer #2: Yes

4. Is the manuscript presented in an intelligible fashion and written in standard English?

PLOS ONE does not copy or edit accepted manuscripts, so the language in submitted articles must be clear, correct, and unambiguous. Any typographical or grammatical errors should be corrected at revision, so please note any specific errors here.

Reviewer #1: Yes

Reviewer #2: Yes

5. Review Comments to the Author

Reviewer #1: This is a valuable contribution to the literature on HIDS (Hyper IgD Syndrome) or MVK deficiency. By presenting a small but by HIDS standard large case series with both clinical and molecular data it is a very useful contribution. The ten patients from six families represent the clinical spectrum previously reported with the additional information on the difficulty making the correct diagnosis and the other disorders that had been entertained before the molecular studies clearly identified the proper defect. Virtually nothing was said about outcome or treatment perhaps because the standard therapies that block IL1 (anakinra and canakinumab) are apparently unavailable or they plan to write another paper with that data. Surprisingly the authors state that the patients did well whereas untreated HIDS is typically associated with poor growth and development. None of the patients had the more severe form of the disease called mevalonic aciduria which is associated with severe and life threatening manifestations in infancy but not the periodic fever syndrome that HIDS displays.

Response: We thank the reviewer for the suggestion. We have included more information regarding the clinical features, treatment, infection, previous diagnosis, development delay, deaths etc. for all the patients in the section on results and discussed them. The clinical features with other clinical details are summarized in the Supplementary Table 1. 

I liked tables 1 and 2 but not Figure 1 which I would like to replace with a linear model of the gene with the mutation sites indicated. Another figure which would be useful is a three dimensional model of the enzyme with the mutation sites identified. A little more focussed discussion of the deletion mutation and its effect on enzyme function or enzyme stability would have been interesting. The sequencing data is consistent with autosomal recessive inheritance requiring either homozygosity or compound heterozygote suggesting single mutations even if the deletion variant have enough residual activity to not be clinically symptomatic.

Response: We thank the reviewer for their suggestion. We have included both the figures i.e. linear model of the gene with the mutation sites indicated. Also 3D model of the enzyme with mutation sites identified using PyMOL in Fig 1. We put the Sanger sequencing with pedigree figure in the supplementary figures. We have also discussed the deletion mutation effect on protein activity and stability in the Discussion part.

Reviewer #2: Thank you for the opportunity to review this manuscript. Authors have described the clinical and molecular spectrum of Hyperimmunoglobulin D Syndrome. I have few comments-

Introduction-

Second para- 4th line- Authors could add more Indian references on MVK disorder. Some of the important references are as follows-

1: Correa ARE, Gupta N, Bagri N, Vignesh P, Alam S, Yamaguchi S. Mevalonate

Kinase Deficiency as Cause of Periodic Fever in Two Siblings. Indian Pediatr.

2020 Feb 15;57(2):180-181. PubMed PMID: 32060250

2: Jain A, Misra DP, Sharma A, Wakhlu A, Agarwal V, Negi VS. Vasculitis and

vasculitis-like manifestations in monogenic autoinflammatory syndromes. Rheumatol

Int. 2018 Jan;38(1):13-24. doi: 10.1007/s00296-017-3839-6. Epub 2017 Oct 14.

Review. PubMed PMID: 29032440.

3: Abdulla MC, Hawkins PN, Alungal J, Rowczenio D. Hyperimmunoglobulin D syndrome

in an Indian family undiagnosed for 11 years. Int J Rheum Dis. 2017

Dec;20(12):2236-2237. doi: 10.1111/1756-185X.12811. Epub 2015 Dec 1. PubMed PMID:

26620804.

4: Sinha A, Waterham HR, Sreedhar KV, Jain V. Novel mutations causing

hyperimmunoglobulin D and periodic fever syndrome. Indian Pediatr. 2012

Jul;49(7):583-5. PubMed PMID: 22885443.

Response: We thank the reviewer for the suggestion, we have added papers from India in the manuscript. 

Methods-

Clinical work up- Whether any biochemical assays like GCMS was performed? In how many clinical diagnoses of Hyper IgD was made in the absence of IgD levels.

Response: Unfortunately, GCMS was not done in any of the patients and IgD estimation was not possible in 70% of the children, although IgA levels were elevated in 100% of children. Total IgD was estimated in 2 patients and IgD surface in 1 patient which has been put in the Supplementary Table 1. 

Variant prioritization- Generally a MAF of 1% is used. The reason behind using MAF of 5% may be specified.

Response: We appreciate the reviewer for pointing this out. We totally agree that in various studies, MAF was taken as 1% for variant filtering. But in a recent study by ClinGen experts showed that higher frequency (MAF>0.05) variants in one or more control populations could be pathogenic or likely pathogenic (PMID:31942019 PMID:30311383). So not to miss any variant that could be a pathogenic we kept the MAF>0.05 for variant filtering. 

Table 1- Elevated IgD is seen in patient 1 only.was it performed in one patient? or was it normal in others. I understand that it is not widely available.

Response: As mentioned by the reviewer, total IgD estimation is not widely available in our country. Hence it could only be done only for two patients and Surface IgD could be done only for one patient. 

Information on consanguinity should be added in the table.

Response: We thank the reviewer for the suggestion, We have added it to Supplementary Table 1 where only 1 family is consanguineous with homozygous mutation. 

Highlight the significance of elevated Vit B12

Response: Unfortunately, we could not find any literature regarding elevated B12 levels in HIDS patients. However, in ALPS which is a well known primary immune deficiency disorder with elevated B12 levels, the reason attributed is an increase in the binding protein haptocorrin (PMID:22306884). Further studies will be necessary to ascertain this finding.

Discussion-

Did authors find any genotype phenotype correlation.

Response: We thank the reviewer for the suggestion, we have added the genotype-phenotype correlations in the discussion.

Authors have stated that p.V377I and c.226+2delT are the common mutations in Indian subcontinent. This should be stated after incorporating the complete Indian data.

Response: We thank the reviewer for this suggestion. We agreed that there should be complete Indian data to state the variant as a common mutation in Indian subcontinent. We have changed it to South India since all our patents belong to South India. All the variants which we found from the literature also had South Indian Ancestry.

Was there any analysis done to prove the founder effect?

Response: We are really thankful to the reviewer for that. We have put the GSA data and performed the identity by descent analysis of our two patients which had same recurrent compound heterozygous mutation to prove mutation to be a founder mutation.

I think this line at the end of discussion ("In a patient who manifest ......... for V377I and c.226+2delT in MVK)may not be generalisable as Indian population is very heterogenous and this study mainly included the south Indian population .

Response: We thank the reviewer for the suggestion, we agreed that the mutation could not be generalized to Indian population. So, we have made the changes of the mutation from Indian population to South Indian population. 

General comments-

Minor typos in introduction and discussion 

Response: We have revised the manuscript and corrected typos.

6. PLOS authors have the option to publish the peer review history of their article (what does this mean?). If published, this will include your full peer review and any attached files.

Do you want your identity to be public for this peer review? For information about this choice, including consent withdrawal, please see our Privacy Policy.

Reviewer #1: No

Reviewer #2: No

---

## [Decision Letter · Decision Letter 1]

31 Jul 2020

PONE-D-20-12540R1

Spectrum of clinical features and genetic variants in mevalonate kinase (MVK) gene of South Indian families suffering from Hyperimmunoglobulin D Syndrome

PLOS ONE

Dear Dr. Scaria,

Thank you for submitting your manuscript to PLOS ONE. After careful consideration, we feel that it has merit but does not fully meet PLOS ONE’s publication criteria as it currently stands. Therefore, we invite you to submit a revised version of the manuscript that addresses the points raised during the review process.

Please see the minor comments raised by Reviewer #1.

Please submit your revised manuscript within 4 weeks. If you will need more time than this to complete your revisions, please reply to this message or contact the journal office at plosone@plos.org. Please include the following items when submitting your revised manuscript:

We look forward to receiving your revised manuscript.

Kind regards,

Dror Sharon, PhD

Academic Editor

PLOS ONE

Journal Requirements:

Additional Editor Comments (if provided):

Reviewers' comments:

Reviewer's Responses to Questions

**Comments to the Author**

1. If the authors have adequately addressed your comments raised in a previous round of review and you feel that this manuscript is now acceptable for publication, you may indicate that here to bypass the “Comments to the Author” section, enter your conflict of interest statement in the “Confidential to Editor” section, and submit your "Accept" recommendation.

Reviewer #1: All comments have been addressed

Reviewer #2: All comments have been addressed

2. Is the manuscript technically sound, and do the data support the conclusions?

Reviewer #1: Yes

Reviewer #2: Yes

3. Has the statistical analysis been performed appropriately and rigorously? 

Reviewer #1: N/A

Reviewer #2: N/A

4. Have the authors made all data underlying the findings in their manuscript fully available?

Reviewer #1: Yes

Reviewer #2: Yes

5. Is the manuscript presented in an intelligible fashion and written in standard English?

Reviewer #1: Yes

Reviewer #2: Yes

6. Review Comments to the Author

Reviewer #1: Much improved. Figure 1 is very helpful. Suppliment table 3 probably should be in the manuscript rather than a supplement. P20 paragraph 4 line 4 except not "expect". Supplement 2 column 1 no $ sign

Reviewer #2: (No Response)

7. PLOS authors have the option to publish the peer review history of their article (what does this mean?). If published, this will include your full peer review and any attached files.

Reviewer #1: No

Reviewer #2: No

---

## [Author Response · Author response to Decision Letter 1]

4 Aug 2020

Comments to the Author

1. If the authors have adequately addressed your comments raised in a previous round of review and you feel that this manuscript is now acceptable for publication, you may indicate that here to bypass the “Comments to the Author” section, enter your conflict of interest statement in the “Confidential to Editor” section, and submit your "Accept" recommendation.

Reviewer #1: All comments have been addressed

Reviewer #2: All comments have been addressed

2. Is the manuscript technically sound, and do the data support the conclusions?

Reviewer #1: Yes

Reviewer #2: Yes

3. Has the statistical analysis been performed appropriately and rigorously?

Reviewer #1: N/A

Reviewer #2: N/A

4. Have the authors made all data underlying the findings in their manuscript fully available?

Reviewer #1: Yes

Reviewer #2: Yes

5. Is the manuscript presented in an intelligible fashion and written in standard English?

Reviewer #1: Yes

Reviewer #2: Yes

6. Review Comments to the Author

Reviewer #1: Much improved. Figure 1 is very helpful. Suppliment table 3 probably should be in the manuscript rather than a supplement. P20 paragraph 4 line 4 except not "expect". Supplement 2 column 1 no $ sign

Response: We thank the reviewer for the suggestion. We have added supplementary table 3 to the manuscript as Table 2 and remove it from the supplementary. We apologize for this mistake, we have corrected the word expect to except. We have removed the text in column 1 of supplementary 2.

Reviewer #2: (No Response)

7. PLOS authors have the option to publish the peer review history of their article (what does this mean?). If published, this will include your full peer review and any attached files.

Do you want your identity to be public for this peer review? For information about this choice, including consent withdrawal, please see our Privacy Policy.

Reviewer #1: No

Reviewer #2: No

---

## [Editor Report · Decision Letter 2]

7 Aug 2020

Spectrum of clinical features and genetic variants in mevalonate kinase (MVK) gene of South Indian families suffering from Hyperimmunoglobulin D Syndrome

PONE-D-20-12540R2

Dear Dr. Scaria,

We’re pleased to inform you that your manuscript has been judged scientifically suitable for publication and will be formally accepted for publication once it meets all outstanding technical requirements.

Kind regards,

Dror Sharon, PhD

Academic Editor

PLOS ONE
---

## [Editor Report · Acceptance letter]

11 Aug 2020

PONE-D-20-12540R2 

Spectrum of clinical features and genetic variants in mevalonate kinase (MVK) gene of South Indian families suffering from Hyperimmunoglobulin D Syndrome 

Dear Dr. Scaria:

I'm pleased to inform you that your manuscript has been deemed suitable for publication in PLOS ONE. Congratulations! Your manuscript is now with our production department. 

Kind regards, 

on behalf of

Prof. Dror Sharon 

Academic Editor

PLOS ONE